# TEC Anomalies Detection for Qinghai and Yunnan Earthquakes on 21 May 2021

**Yingbo Yue** [1,2] , **Hannu Koivula** [3], **Mirjam Bilker-Koivula** [3] , **Yuwei Chen** [4] , **Fuchun Chen** [1,*]  **and Guilin Chen** [1]

1. Key Laboratory of Infrared System Detection and Imaging Technology, Shanghai Institute of Technical Physics, Chinese Academy of Sciences, Shanghai 200083, China
2. University of Chinese Academy of Sciences, Beijing 100049, China
3. Department of Geodesy and Geodynamics, Finnish Geospatial Research Institute, 02151 Espoo, Finland
4. Department of Remote Sensing and Photogrammetry, Finnish Geospatial Research Institute, 02151 Espoo, Finland
* Correspondence: fuchun.chen@mail.sitp.ac.cn

**Abstract:** Earthquake prediction by using total electron content is a commonly used seismic research method. The long short-term memory model is a kind of method to predict time series and has been used for the prediction of total electron content, and the relative power spectrum method is one of the pre-seismic infrared anomaly detection algorithms in the frequency domain. In this paper, a new method combining these two algorithms is used to extract abnormal signals; thus scientists can more easily detect anomalies of total electron content similar to those before the Qinghai and Yunnan earthquakes happened on 21 May 2021. There are pre-seismic anomalies with the high-value relative power spectrum near two epicenters. To validate the correlation between anomalies and earthquakes statistically, the spatiotemporal characteristics of TEC anomalies are analyzed based on connected region recognition. Then, the proportion of earthquake-related anomalies (the correlation rate), the proportion of earthquakes outside the predicted range (the miss rate), and the ratio of the proportion of earthquakes within the predicted range to the spatiotemporal occupancy of anomalies, which is called the probability gain, were used to assess the method. The appropriate parameters of the algorithm for the miss rate below 50% were searched. The highest probability gain is 1.91, which means anomalies of total electron content may decrease the uncertainty of earthquake prediction.

**Keywords:** total electron content; earthquake prediction; pre-seismic anomaly; long short-term memory; relative power spectrum

## 1. Introduction

Earthquake prediction is a challenging problem worldwide. During the seismic preparation period, slight deformation and compression of the crust may cause changes in multiple parameters of the earth's surface and atmosphere [1]. Energy flows from the earth's surface to the atmosphere is the basis of earthquake prediction using the remote sensing data from satellites [2]. Freund et al. explained the pre-seismic phenomenon by using the "peroxy defects theory" [3]. Pulinets et al. unified the concepts of different types of pre-seismic anomalies [4]. However, currently, no precise model can explain the physical and chemical process during the seismic preparation period because of the complex and variable nature of the lithosphere–atmosphere–ionosphere coupling system, which may be implemented by several mechanisms [5]. As a result, it is difficult to directly identify earthquake-related information from raw observations [6]. Previous research shows that long-time monitoring and statistical analysis are significant in finding earthquake precursors. Some scholars found pre-seismic anomalies from the data collected by ground-based on-site stations combined with the satellite's remote observation, including changes

in temperature, humidity, air components (such as $CH_4$, $CO_2$ [7], $O_3$, and so on), and electromagnetic field, etc. [2].

Total electron content (TEC) is one of the important parameters of the ionosphere. TEC is the total number of electrons integrated between two points, and it is significant in determining the scintillation, the group and phase delays of a radio wave through a medium [8]. Thus, ionospheric TEC is characterized by observing carrier phase delays of received radio signals transmitted from satellites located above the ionosphere, such as the global navigation satellites system (GNSS) [9]. Furthermore, it is well-known that TEC is strongly affected by solar activity, and normally the unit of TEC is TECu, 1 TECu = $10^{16}$ electrons/m$^2$. Previous research observed the TEC anomalies before the outbreak of several earthquakes. Kuo et al. proposed an improved model to explain the TEC anomalies before strong earthquakes [10]. However, several factors may affect the TEC, such as seasonal change, solar activity, etc. Therefore, reliable detection methods are necessary because it is difficult to distinguish directly the TEC change caused by seismic and non-seismic factors. Therefore, many algorithms have been developed to detect earthquake-related TEC anomalies, such as median and inter-quartile range [11–14], wavelet transformation [15], Kalman filter, support vector machines [16], neural network [17–19], and the genetic algorithm [20]. These algorithms may be classified into three categories, including statistical methods, frequency-domain analysis, and time series predictions.

The long short-term memory (LSTM) network is a common recursive neural network that has the advantage of predicting series because of its memorability and shared parameters. It has been used to predict the time series of TEC with better performance (in terms of the coefficient of determination, root mean square error values, mean absolute error, and correlation coefficient) than the deep neural network, autoregressive integrated moving average (ARIMA) model and international reference ionosphere (IRI) model [21]. A previous study compares LSTM and ARIMA for TEC anomalies detection before the Haiti earthquake [22]. It shows that LSTM has better performance than ARIMA because of the lower mean square error (MSE) and root means square error (RMSE). The two methods could predict the normal TEC value by using previous data. Akhoondzadeh et al. [14,23] used the LSTM model to extract the TEC anomalies from the Swarm satellite. The relative power spectrum method transforms time series into the frequency domain and has been used to detect infrared anomalies before earthquakes [24–26]. It can detect some anomalies that are difficult to be observed in the time domain, by extracting the changes in the power spectrum at the characteristic frequency. To the knowledge of the authors, there is no research about the pre-seismic TEC anomaly detected with the relative power spectrum. Long-term statistics are necessary for the validation of the correlation between TEC anomalies and earthquakes. Some previous studies used the receiver operating characteristic (ROC) curve [13] and the Molchan diagram [27,28]. However, the number of non-seismic cases is far larger than that of seismic cases in the real world. The data imbalance and other problems cause the ROC to not be suitable for earthquake prediction [29,30].

In this paper, the relative power spectrum is used to analyze the deviation of the LSTM model, extracting the abnormal signal to make it more obvious. Most relevant studies only used short-time TEC data around the time of the earthquake occurrence [31,32]. Some statistical researches show that TEC anomalies could be detected before most strong earthquakes [33–35]. However, the earthquake precursor is not credible without calculating the probability of an earthquake occurrence after TEC anomaly observation. In this paper, four-year (from 2016 to 2019) data are used to train the model. The background (annual trend and seasonal change) has better visibility with a larger dataset. The abnormal signals were detected by the relative power spectrum after removing the background. The TEC anomalies for two years (from 2020 to 2021) are analyzed statistically to reveal the correlation between the TEC anomalies and earthquakes.

## 2. Materials and Methods

### 2.1. Data and Study Area

Madrigal is an upper atmospheric science database (http://madrigal.iggcas.ac.cn/ (accessed on 14 January 2022)). It has collected the TEC data from the worldwide GNSS receiver network since 1998. Every file provides the one-day data as a three-dimension grid. The spatial resolution of the Madrigal dataset is $1° \times 1°$, and the temporal resolution is five minutes [36]. This paper uses the average hourly TEC to construct a new time series.

China is a country in which earthquakes happen frequently. The Qinghai earthquake, with a magnitude of 7.4 on 21 May 2021 (UTC), was the strongest earthquake during the last decade in China. Such an event is known also as a "Madoi earthquake" [37]. The epicenter was located at 98.34°E, 34.59°N. It was in a high-mountain context, luckily, with a small population and no casualties reported [38]. In this paper, six years of TEC data was applied to the LSTM model. About two-thirds of them (from 2016 to 2019) are used to construct the training dataset, whereas the other data (from 2020 to 2021) are used to construct the test dataset and study the abnormal phenomenon. The epicenters of the earthquakes in China and the surrounding area from 2020 to 2021 are shown in Figure 1. The TEC data are used to study the earthquakes within the green frame (from 90°E to 110°E, and from 17°N to 42°N). A total of 31 earthquakes with a magnitude of 5 or greater occurred in this region from 2020 to 2021. The earthquake information is in Table 1, including the earthquake time (UTC), the magnitude and the epicenter positions. The distribution of their depth is shown in Figure 2. The deepest earthquake occurred at a depth of about 100 km. Most earthquakes are shallow-focus earthquakes the depths of which are no more than 60 kilometers. The magnitudes and times of these earthquakes are shown in Figure 3. Four M5+ earthquakes happened in Yunnan within 5 h before the M7.4 Qinghai earthquake, suggesting a possible seismic trigger of this larger event. In addition, one aftershock with a magnitude of 5.1 happened in Qinghai about eight hours after the M7.4 Qinghai earthquake. The M7.4 Qinghai earthquake and M6.4 Yunnan earthquake are regarded as the mainshocks. Their magnitudes are larger than that of the foreshocks and the aftershocks, which happened in their neighborhood during a short period.

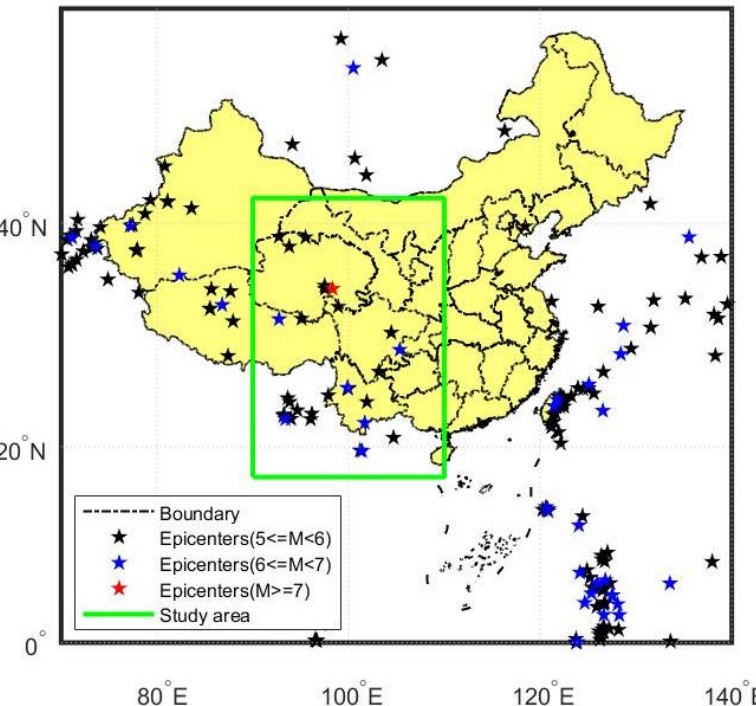

**Figure 1.** The epicenters in China and the surrounding area from 2020 to 2021.

**Table 1.** The information list of the earthquakes occurred in the study area from 2020 to 2021.

| Earthquake Time (UTC) | Magnitude | Latitude (°E) | Longitude (°N) | Depth (km) | Type |
|---|---|---|---|---|---|
| 24 January 2020 22:56:05 | 5.1 | 31.98 | 95.09 | 10 | Mainshock |
| 2 February 2020 16:05:41 | 5.1 | 30.74 | 104.46 | 21 | Mainshock |
| 1 April 2020 12:23:27 | 5.6 | 33.04 | 98.92 | 10 | Mainshock |
| 16 April 2020 11:45:25 | 5.8 | 22.72 | 94.00 | 10 | Mainshock |
| 18 May 2020 13:47:59 | 5.0 | 27.18 | 103.16 | 8 | Mainshock |
| 25 May 2020 14:42:16 | 5.1 | 24.35 | 93.85 | 60 | Mainshock |
| 21 June 2020 22:40:53 | 5.7 | 23.15 | 93.25 | 30 | Mainshock |
| 27 July 2020 05:14:48 | 5.3 | 20.90 | 104.70 | 10 | Mainshock |
| 27 August 2020 12:07:15 | 5.2 | 23.00 | 93.20 | 10 | Mainshock |
| 10 October 2020 17:38:00 | 5.1 | 24.70 | 93.65 | 40 | Mainshock |
| 14 November 2020 08:50:26 | 5.2 | 23.55 | 94.65 | 100 | Mainshock |
| 19 March 2021 06:11:26 | 6.1 | 31.94 | 92.74 | 10 | Mainshock |
| 21 May 2021 13:21:25 | 5.6 | 25.63 | 99.92 | 10 | Foreshock |
| 21 May 2021 13:48:34 | 6.4 | 25.67 | 99.87 | 8 | Mainshock |
| 21 May 2021 13:55:28 | 5.0 | 25.67 | 99.89 | 8 | Aftershock |
| 21 May 2021 14:31:10 | 5.2 | 25.59 | 99.97 | 8 | Aftershock |
| 21 May 2021 18:04:11 | 7.4 | 34.59 | 98.34 | 17 | Mainshock |
| 22 May 2021 02:29:34 | 5.1 | 34.85 | 97.50 | 10 | Aftershock |
| 10 June 2021 11:46:07 | 5.1 | 24.34 | 101.91 | 8 | Mainshock |
| 12 June 2021 10:00:46 | 5.0 | 24.96 | 97.89 | 16 | Mainshock |
| 16 June 2021 16:48:58 | 5.8 | 38.14 | 93.81 | 10 | Mainshock |
| 7 July 2021 14:43:48 | 5.2 | 19.65 | 101.20 | 10 | Mainshock |
| 29 July 2021 16:39:27 | 5.7 | 22.70 | 96.04 | 20 | Mainshock |
| 13 August 2021 12:21:35 | 5.8 | 34.58 | 97.54 | 8 | Mainshock |
| 26 August 2021 07:38:18 | 5.5 | 38.88 | 95.50 | 15 | Mainshock |
| 16 September 2021 04:33:31 | 6.0 | 29.20 | 105.34 | 10 | Mainshock |
| 26 November 2021 07:45:42 | 6.1 | 22.70 | 93.40 | 50 | Mainshock |
| 6 December 2021 08:25:38 | 5.0 | 23.23 | 96.17 | 10 | Mainshock |
| 19 December 2021 07:54:28 | 5.3 | 38.95 | 92.73 | 10 | Mainshock |
| 20 December 2021 05:06:14 | 6.0 | 19.60 | 101.40 | 10 | Mainshock |
| 24 December 2021 21:43:21 | 6.0 | 22.33 | 101.69 | 15 | Mainshock |

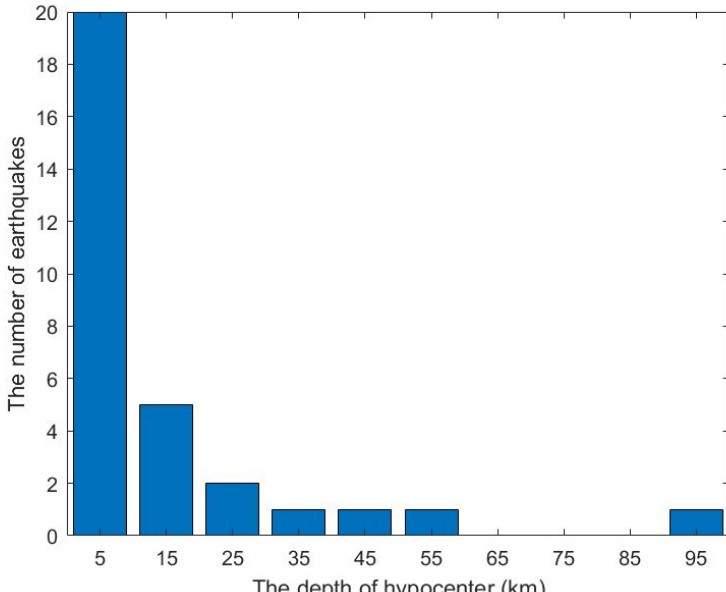

**Figure 2.** The distribution of hypocenter depths.

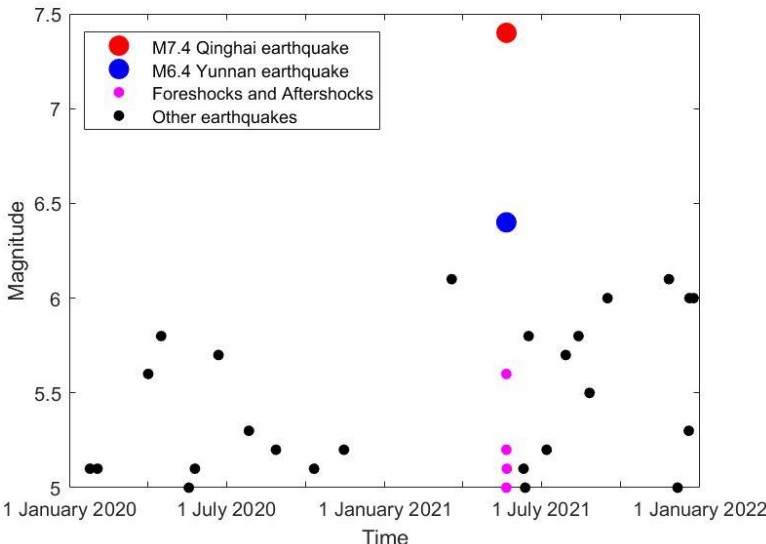

**Figure 3.** The time and magnitude of the earthquakes.

The grid data collected from Madrigal could only cover a partial area due to the limitation of GNSS receivers. Some points without the measured value of TEC are marked as not a number (NaN) in the grid data. Figure 4 shows the percentage of points with the value of NaN among the time series in every position. Only the data on the northwest side of the epicenters could be used for the research of the Qinghai earthquake because most points that are on the other side are not covered.

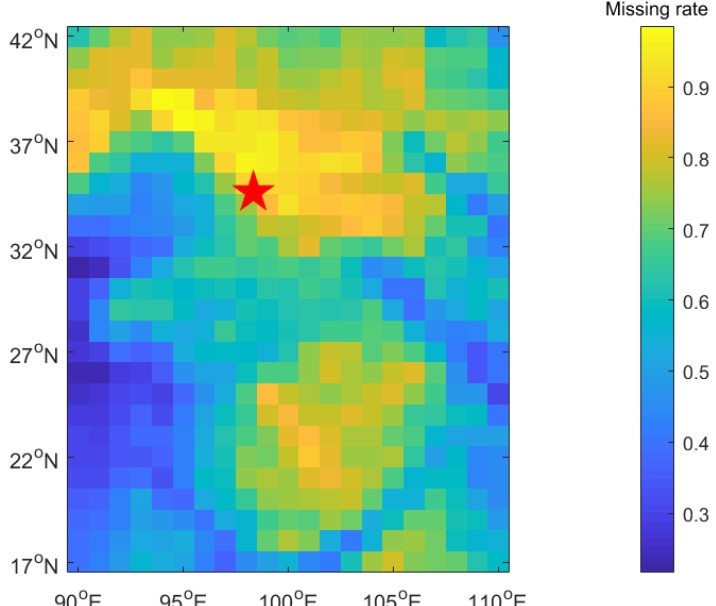

**Figure 4.** The percentage of invalid data.

### 2.2. Anomalies Detection

#### 2.2.1. LSTM Model

The hourly TEC data from 2016 to 2021 are used to construct the training and test dataset. For a time series ($D = \{d_1, d_2, \ldots, d_N\}$) with the length of N, the first m data in series D are used to build the training set, and the rest are used to build the test set. The training pattern is shown in Equations (1)–(3),

$$y_t = f(x_t) \tag{1}$$

$$y_t = [d_t] \tag{2}$$

$$x_t = [d_{t-\tau s}, d_{t-(\tau-1)s}, \dots, d_{t-s}], \tag{3}$$

where $x_t$ is the input vector, $y_t$ is the label, $\tau$ is the input length, s is the sampling time spacing, and $t = \tau s + 1$, $\tau s + 2$, $\dots$, $m$. In this paper, $\tau = 3$, s = 24, N = 52608, m = 0.67·N ≈ 35247. In other words, one value may be predicted by using the data of previously 72 h, 48 h and 24 h. The test pattern is similar to the training pattern, but $t = m + \tau s + 1$, $m + \tau s + 2$, $\dots$, N. The structure of the LSTM layer is shown in Figure 5. The state information of the LSTM at the moment could be transferred to that at the next moment. Equations (4)–(9) show the specific transfer functions [39],

$$f_t = \sigma\left(W_f \cdot [h_{t-1}, x_t] + b_f\right) \tag{4}$$

$$i_t = \sigma(W_i \cdot [h_{t-1}, x_t] + b_i) \tag{5}$$

$$\widetilde{C}_t = tanh(W_c \cdot [h_{t-1}, x_t] + b_c) \tag{6}$$

$$C_t = f_t * C_{t-1} + x_t * \widetilde{C}_t \tag{7}$$

$$O_t = \sigma(W_o \cdot [h_{t-1}, x_t] + b_o) \tag{8}$$

$$h_t = O_t * \tan h(C_t), \tag{9}$$

where $h_{t-1}$ is the output of the last state, $C_{t-1}$ is the transferred information of the last state, $\sigma$ is a sigmoid neural network layer, tanh in Equation (6) is a tanh neural network layer, tanh in Equation (9) is a tanh function, $h_t$ is the output at the moment t, $C_t$ is the state information at the moment t and will be transferred to the next state, and $W_f$, $W_i$, $W_c$, $W_o$, $b_f$, $b_i$, $b_c$, $b_o$ are the parameters of the LSTM layer.

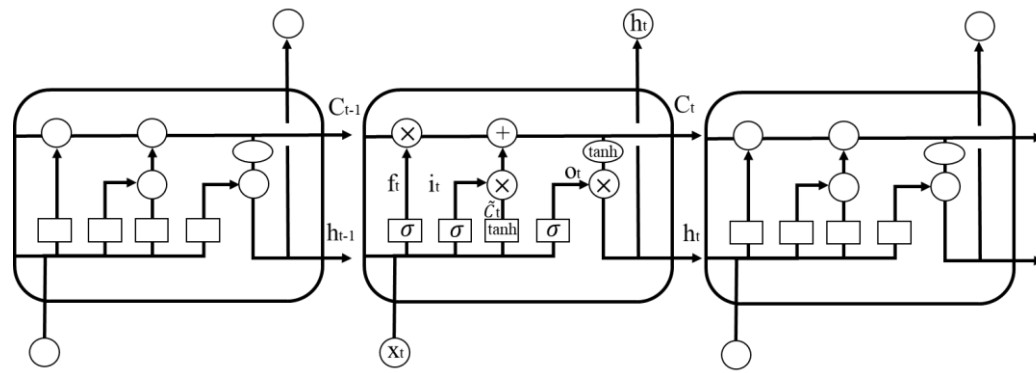

**Figure 5.** The structure of the LSTM layer.

The fully connected layer could transform the output of the LSTM layer into the same dimension as the label, as shown in Equation (10) [40],

$$\hat{y}_t = W_d \cdot h_t + b_d, \tag{10}$$

where $W_d$, $b_d$ are the parameters of the fully connected layer and $\hat{y}_t$ is the output of the model and the predicted value at moment t. The root means square error (RMSE) could evaluate the model's performance. Its definition is shown in Equations (11) and (12) [41],

$$RMSE_{train} = \sqrt{\frac{1}{m - \tau s}\sum_{k=\tau s+1}^{m}(\hat{y}_k - y_k)^2} \tag{11}$$

$$RMSE_{test} = \sqrt{\frac{1}{N-m-\tau s}\Sigma_{k=m+\tau s+1}^{N}(\hat{y}_k - y_k)^2}, \tag{12}$$

where $RMSE_{train}$ is the RMSE of the training dataset and $RMSE_{test}$ is the RMSE of the test dataset.

### 2.2.2. Relative Power Spectrum

The difference between the observed value ($y_k$) and the predicted value ($\hat{y}_k$) may construct a new error series (E = $\{e_1, e_2, \ldots, e_M\}$, $e_k = y_k - \hat{y}_k$), where M is the length of the data. The data from 2020 to 2021 is used to analyze the pre-seismic anomalies. A relative power spectrum is used to analyze the series. For a given window length ($\varepsilon$) and step length (p), the series could be resampled into multiple series, as Equation (13) shows:

$$g_t = [e_{t-\varepsilon+1}, e_{t-\varepsilon+2}, \ldots, e_{t-1}, e_t], \text{ t} = \varepsilon,\ \varepsilon+\text{p}, \varepsilon+2\text{p}, \ldots, \text{M}. \tag{13}$$

Every resampled series are transformed into a frequency domain by discrete Fourier transform, as Equation (14) shows [42],

$$f_t(k) = \Sigma_{n=1}^{\varepsilon} g_t(n)e^{-j\frac{2nk\pi}{\varepsilon}},\ k = 0,\ 1,\ 2,\ \ldots,\ \varepsilon-1, \tag{14}$$

where $g_t(n)$ is the nth value in the series $g_t$; k is the independent variable of the spectrum. The relative power spectrum is defined in Equation (15) [25],

$$R_t(k) = \frac{f_t(k)\cdot\overline{f}_t(k)}{\frac{p}{M-\varepsilon+p}\Sigma_{t=\varepsilon,\varepsilon+\text{p},\varepsilon+2\text{p},\ldots,M} f_t(k)\cdot\overline{f}_t(k)}, \tag{15}$$

where $\overline{f}_t(k)$ is the conjugate complex of $f_t(k)$. If $R_{t_0}(k_0) \geq R_t(k)$ for any $t \in \{\varepsilon,\ \varepsilon+\text{p},\ \varepsilon+2\text{p},\ \ldots,\ \text{M}\}$ and $k \in \{0, 1, 2, \ldots, \varepsilon-1\}$, $R_t(k_0)$ is defined as the relative power spectrum series at the characteristic frequency. In this paper, $M = 17544$, $p = 24$, and $\varepsilon = 1536$. These parameters (one-day step length and 64-day window length) are determined by the previous algorithm for infrared data [25].

### 2.3. Statistical Method

The pre-seismic infrared anomaly with a high value of relative power spectrum has been explored in previous research [26,42]. Similarly, the point with a high value of TEC relative power spectrum may be regarded as a possible pre-seismic anomaly, but it needs statistical validation. The value of the relative power spectrum indicates no anomaly or the intensity of the anomaly. For a given intensity threshold ($T_i$), the points with a value higher than $T_i$ are defined as abnormal points. To study the spatiotemporal characteristics and the correlation between anomalies and earthquakes, the abnormal points in the same connected region belong to one anomaly. As Figure 6 shows, the neighborhood of the red point P includes 26 points (blue points). For one abnormal point, if another point in its neighborhood is also abnormal, these two points are in the same connected region and belong to the same anomaly. For one anomaly including z points (A = $\{P_1(\alpha_1, \beta_1, \gamma_1), P_2(\alpha_2, \beta_2, \gamma_2), \ldots, P_1(\alpha_z, \beta_z, \gamma_z)\}$), where $\alpha_k, \beta_k, \gamma_k$ are longitude, latitude, and time of the kth point $P_k$. The duration (T) is defined in Equation (16),

$$\text{T} = \max\{\gamma_1, \gamma_2, \ldots, \gamma_z\} - \min\{\gamma_1, \gamma_2, \ldots, \gamma_z + 1\}. \tag{16}$$

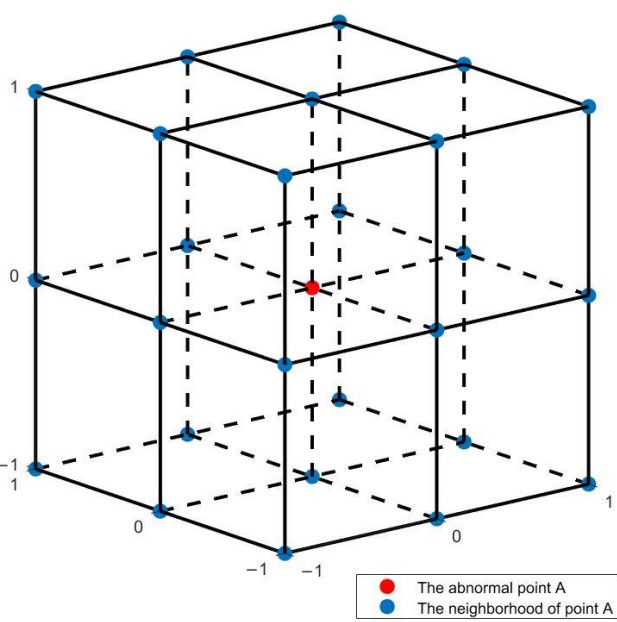

**Figure 6.** The schematic diagram of the neighborhood region.

The covering area (S) is defined in Equations (17) and (18),

$$S = \sum_{lat=\beta_S}^{\beta_N} \sum_{lon=\alpha_W}^{\alpha_E} h(lon, lat) \tag{17}$$

$$h(lon, lat) = \begin{cases} 1, \ if \ \exists k \in \{c_m, \ c_m + 1, \ \ldots, \ c_M\}, \ point(lon, \ lat, \ k) \in A \\ 0, \ if \ \forall k \in \{c_m, \ c_m + 1, \ \ldots, \ c_M\}, \ point(lon, \ lat, k) \notin A' \end{cases} \tag{18}$$

where $\alpha_W$, $\alpha_E$ are the minimum and maximum longitude in the study region, $\beta_S$, $\beta_N$ are the minimum and maximum latitude in the study region, and $c_m$, $c_M$ are the minimum and maximum of the studied period.

For a given predicted radius (R) and predicted time window (W), one point belongs to the predicted range if the spatial distance from it to any point in set A is less than R and the point is within W days from the earliest time in set A. The anomaly is regarded as a pre-seismic anomaly if any earthquake with a magnitude over 5 occurred in the predicted range determined by it. According to the statistical analysis (confusion matrix) of TIR anomalies by Zhang et al., the correlation rate (CR) is the proportion of the pre-seismic anomalies among all anomalies, as Equation (19) shows [27],

$$CR = \frac{N_P}{N_A}, \tag{19}$$

where $N_P$ is the number of pre-seismic anomalies and $N_A$ is the number of all anomalies. According to the method for testing alarm-based earthquake predictions proposed by Zechar et al., the miss rate of earthquake prediction (ν) is the proportion of the earthquakes outside the predicted range among all earthquakes, as Equation (20) shows [29],

$$N = \frac{N_E - N_T}{N_E}, \tag{20}$$

where $N_T$ is the number of earthquakes following pre-seismic anomalies and $N_E$ is the number of all target earthquakes. In this paper, only the mainshocks with an epicenter depth less than or equal to 60 kilometers are considered, so $N_E$ is equal to 26. The probability gain (G) is shown in Equation (21) [28,29],

$$G = \frac{1 - \nu}{\rho},\tag{21}$$

where $\rho$ is the alarming rate [43], which is the proportion of the number of pixels in the predicted range to that in the whole study range. Large R and W may increase CR and decrease $\nu$ simultaneously but lead to a high spatiotemporal occupation rate ($\rho$). The two parameters should be set neither too high nor too low to avoid a small probability gain. The prediction performance could be improved by selecting appropriate R and W.

## 3. Results and Discussion

### 3.1. LSTM Model Performance

Approximately two-thirds of the TEC data are used to train the model, and the rest are used for testing. The RMSE of the training dataset and the test dataset are shown in Figure 7. The RMSE of most points in the northern region is lower than that in the southern region. The maximum train RMSE is 5.9, while that of test RMSE is 6.5. Comparing Figures 4 and 7, there is no RMSE in the region with high data missing rate, because the valid data is too little to generate a dataset. The frequency distribution of test RMSE in the valid region is shown in Figure 8. The test RMSE at 97.85% of valid points is less than five.

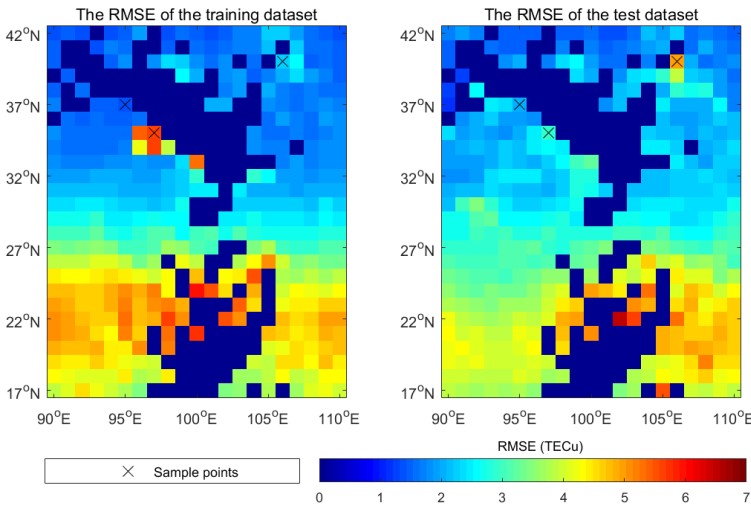

**Figure 7.** The RMSE of the training and test dataset.

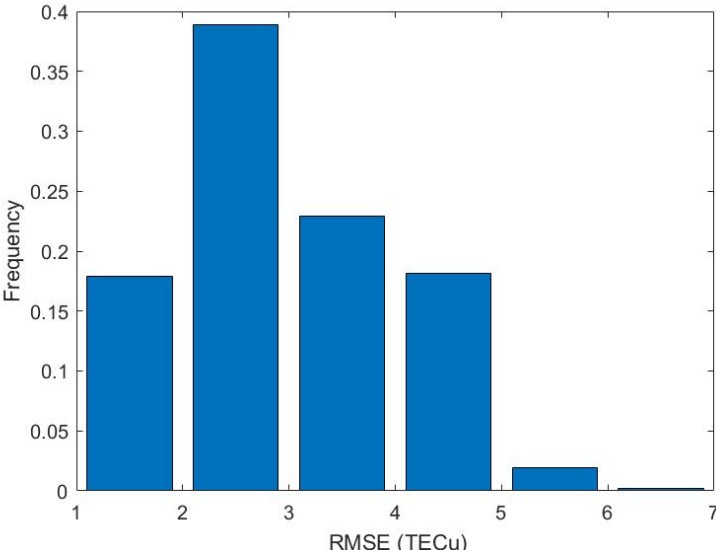

**Figure 8.** The frequency distribution of test RMSE.

Figure 9 shows the observed TEC and predicted TEC in three positions. The RMSEs are shown in Table 2. There are low Train RMSE and high Test RMSE at 106°E, 40°N, whereas there are high Train RMSE and low Test RMSE at 97°E, 35°N. The Train RMSE and Test RMSE are both low at 95°E, 37°N. As Figure 9 shows, the model's prediction performance may be poor when the TEC increases, as expected. The error at 95°E, 37°N, which is the deviation of observed TEC from the predicted TEC, is shown in Figure 10 (black line). It fluctuates around zero. The Dst index (blue line) and the Kp index (orange line) may reveal the intensity levels of geomagnetic activity and have been testified to be able to influence the TEC [44]. The two kinds of geomagnetic indices could be collected on the Madrigal website. Generally, geomagnetic activity is quiet when the Kp is less than 3. Larger Kp means more strong geomagnetic activity. The geomagnetic activity is quiet as Dst is closer to zero. When a storm occurs, the Dst rises transitorily followed by a sharp fall [45]. Figure 10 shows that there is no obvious correlation between the error and the geomagnetic activity. Figure 11 shows the frequency distribution of the error at 95°E, 37°N. About 78% of the errors were between −2 TECu and 2 TECu.

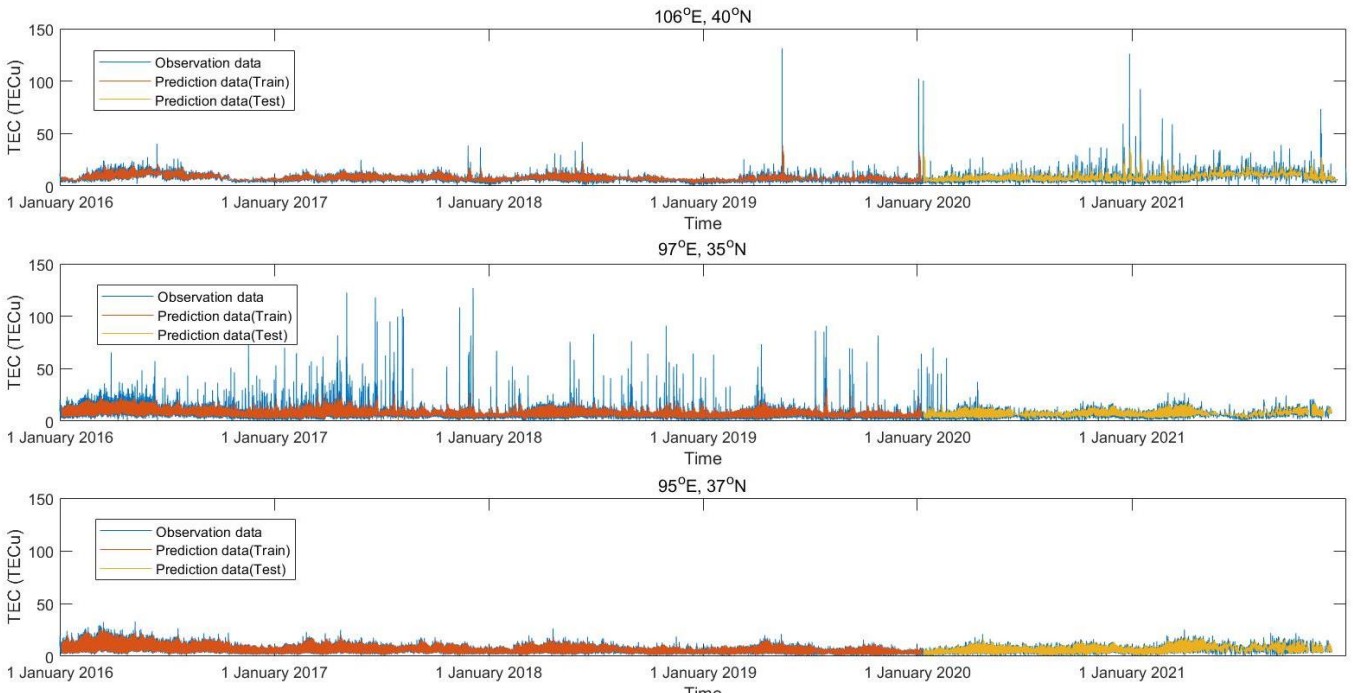

**Figure 9.** The observation and prediction data using LSTM model of three positions.

**Table 2.** The train RMSE and test RMSE at three positions.

| Position | 106°E, 40°N | 97°E, 35°N | 95°E, 37°N |
|---|---|---|---|
| Train RMSE | 2.2807 | 5.6363 | 1.3819 |
| Test RMSE | 5.0072 | 2.9243 | 2.0781 |

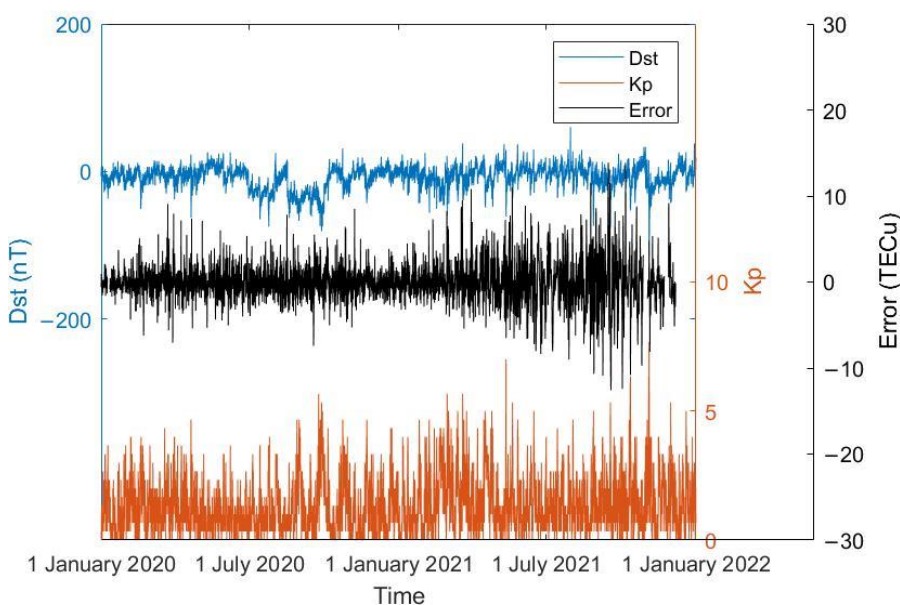

**Figure 10.** The deviation of observed TEC from predicted TEC.

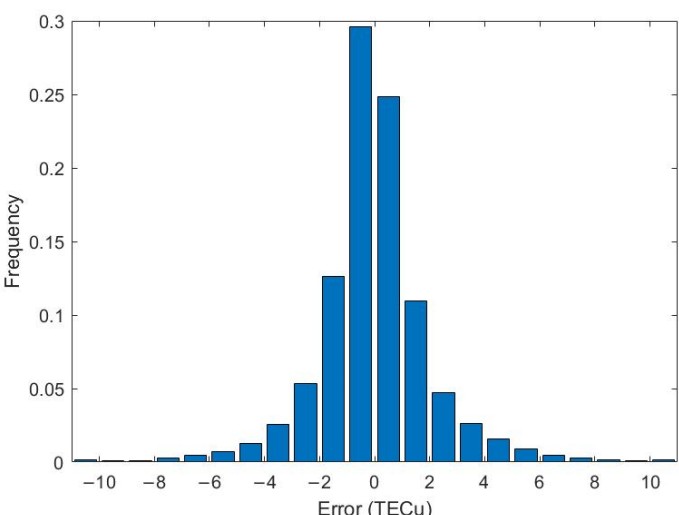

**Figure 11.** The frequency distribution of the error.

### 3.2. Statistic Result

There are five significant parameters for earthquake prediction using the relative power spectrum, including intensity threshold, duration threshold, area threshold, predicted time window, and predicted radius. The intensity threshold is the criterion of anomaly identification. The duration and area depict the spatiotemporal occupation range of an anomaly. The anomaly that lasts no less than $T_d$ and covers no less than $T_a$ is used to predict earthquakes. The predicted time window and predicted radius determine the predicted range. A large predicted range means it is difficult to predict the exact time and position of the earthquake. In this paper, two-year anomalies are analyzed statistically with different parameters. The value range of these parameters is shown in Table 3. They determine the performance of earthquake prediction.

**Table 3.** The value of algorithm parameters.

| Parameters | Intensity Threshold $T_i$ | Duration Threshold ($T_d$, Days) | Area Threshold ($T_a$, Pixels) | Predicted Time Window (W, Days) | Predicted Radius (R, km) |
|---|---|---|---|---|---|
| Value | 5, 10, 15, 20 | 1, 2, 3, 4, 5, 6, 7 | 1, 2, 3, 4, 5 | 14, 28, 42, 56 | 100, 200, 300, 400, 500 |

The CR, $\nu$ and G are calculated with multiple groups of parameters. The distribution of these three statistical results is shown in Figure 12. The G could be used to prove that the earthquake precursors are not occasional. Many points with a high G are above the black line, but their miss rates are too high to predict a reasonable number of earthquakes. Effective earthquake precursors need a low miss rate, which means that they can be detected in advance of most earthquakes. Table 4 shows the top ten groups with the highest gains among the groups with a TPR above 50%, which are ordered by decreasing G. The G of group 1 is 1.91, which is far larger than that of other groups. With the parameters in group 1 ($T_i = 5$, $T_d = 3$ days, $T_a = 2$ pixels, W = 28 days, R = 300 km), 10% of anomalies are earthquake-related, whereas 40% of earthquakes are outside the predicted range. The points of group 1 (the dark red point) and group 2 (the bright red point) are easy to identify below the black line in Figure 12 because all points with G below 1.5 are marked as the black points.

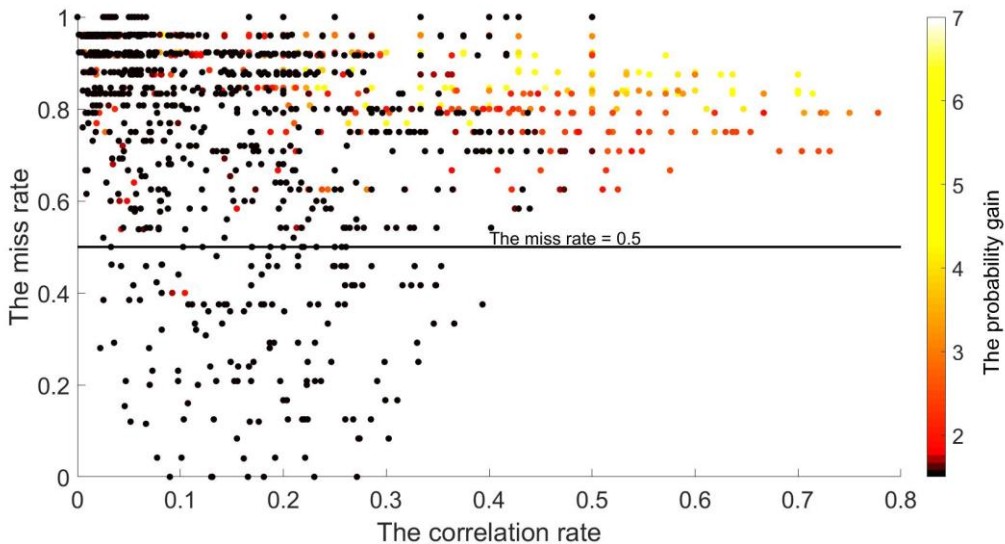

**Figure 12.** The distribution of the correlation rate, the miss rate and the probability gain.

**Table 4.** The statistical evaluation of the algorithm with different parameters.

| Group Number | $T_v$ | $T_d$ (days) | $T_a$ (pixels) | W (days) | R (km) | CR | $\nu$ | G |
|---|---|---|---|---|---|---|---|---|
| 1 | 5 | 3 | 2 | 28 | 300 | 0.10 | 0.40 | 1.91 |
| 2 | 5 | 2 | 2 | 28 | 300 | 0.10 | 0.40 | 1.68 |
| 3 | 15 | 1 | 1 | 14 | 400 | 0.08 | 0.46 | 1.59 |
| 4 | 5 | 6 | 2 | 42 | 400 | 0.22 | 0.50 | 1.58 |
| 5 | 5 | 6 | 2 | 56 | 400 | 0.27 | 0.42 | 1.58 |
| 6 | 5 | 6 | 1 | 56 | 300 | 0.19 | 0.50 | 1.58 |
| 7 | 15 | 1 | 1 | 14 | 500 | 0.12 | 0.31 | 1.57 |
| 8 | 5 | 3 | 2 | 42 | 300 | 0.14 | 0.38 | 1.56 |
| 9 | 5 | 1 | 2 | 28 | 300 | 0.08 | 0.40 | 1.56 |
| 10 | 15 | 1 | 1 | 28 | 300 | 0.10 | 0.44 | 1.55 |

### 3.3. Cases Study

There are six earthquakes in the study region with a magnitude over 5 from 21 May to 22 May in 2021. The specific information is shown in Table 4. The former four earthquakes happened in Yunnan province with a maximum magnitude of 6.4. The latter two earthquakes occurred in Qinghai province with a maximum magnitude of 7.4. Based on the statistical results in Section 3.2, the best-predicted radius is 300 km, which is the predicted radius of group 1 in Table 4. Abnormal points within 300 km of the epicenters are shown in Figures 13 and 14. The horizontal axis is the number of days of the anomaly time relative to earthquake time, and the negative number is before the earthquake. The vertical axis is the distance from the anomaly to the epicenter. The anomaly intensity are the value of the relative power spectrum of no less than five.

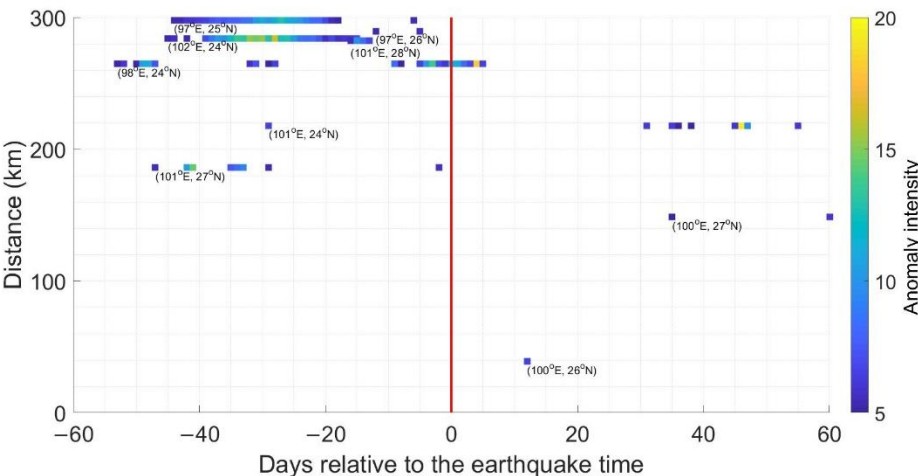

**Figure 13.** The anomalies around the Yunnan epicenter.

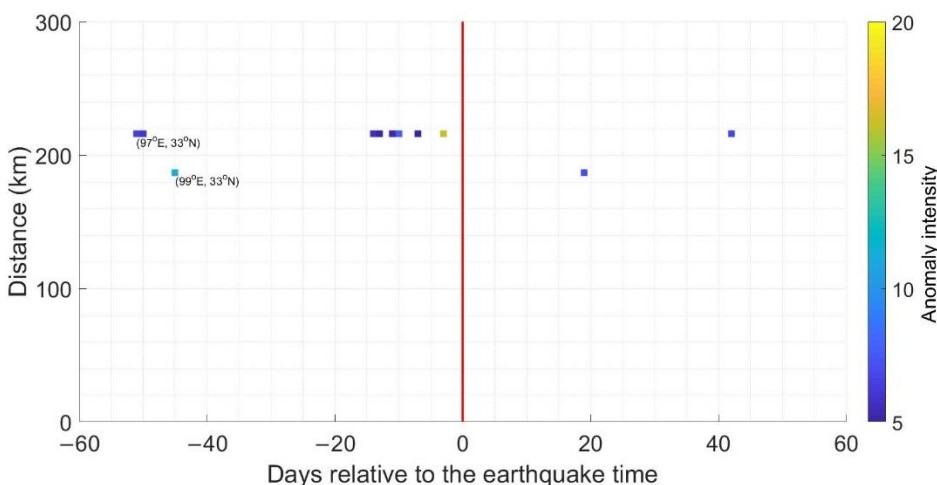

**Figure 14.** The anomalies around the Qinghai epicenter.

Figure 13 shows the abnormal points around the Yunnan earthquake with a magnitude of 6.4. There were seven abnormal points around the epicenter within 60 days before the earthquake. The continuous anomalies lasted for more than seven days at three of these points. For example, the relative power spectrum appeared anomaly five days before the earthquake and remained until five days after the earthquake at 98°E, 24°N. Figure 14 shows the anomalies before the Qinghai earthquake with a magnitude of 7.4, including two abnormal points. Two anomalies were at 99°E, 33°N. One is before the earthquake, and the other one is after the earthquake. There were intermittent anomalies within two

weeks before the earthquake at 97°E, 33°N, and the maximum occurred three days before the earthquake. To compare the anomalies of nine abnormal points, their time curve is rendered in Figure 15. The picture on the bottom is the magnification of the partial upper picture. The solid black line represents the 28 days before the earthquake, which is the predicted time window of group 1 in Tabel 4. In both cases, anomalies were found near the epicenter within 28 days before the earthquakes.

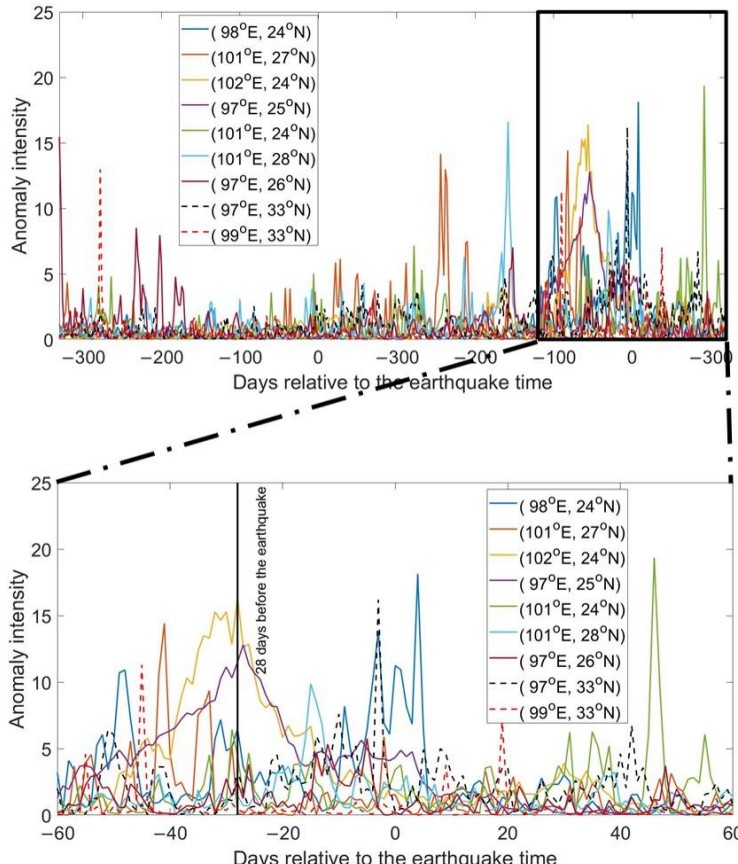

**Figure 15.** The time curves of abnormal values.

The most obvious anomaly is at 102°E, 24°N. It lasted for 25 days and had the highest peak among all pre-seismic anomalies. Significant anomalies were observed near both two epicenters three days before the earthquakes. (The dark blue solid line and the black dotted line). The relative power spectrum on that day is shown in Figure 16. The anomaly value around the Qinghai epicenters is slightly higher than that around the Yunnan epicenters. It is in accordance with the magnitude of the two earthquakes. The strongest pre-seismic anomaly around the Yunnan epicenter occurred 28 days before the earthquake. As Figure 17 shows, a large-area anomaly appeared in the southeast direction of the epicenter, and the closest distance to the epicenter was less than 300 km.

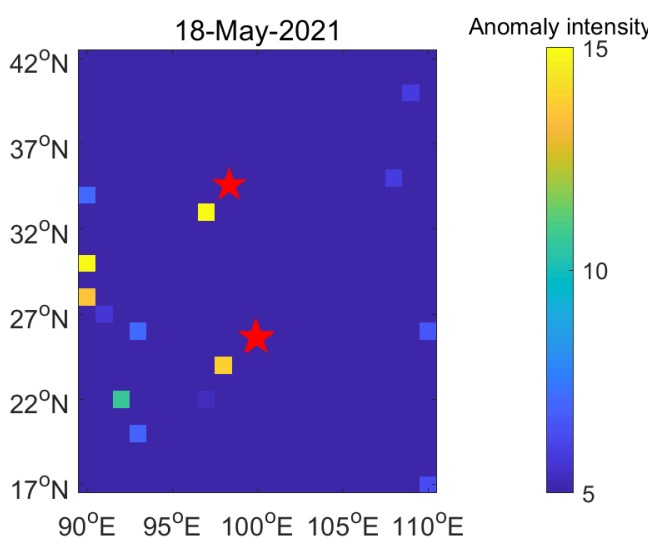

**Figure 16.** The relative power spectrum on 18 May 2021.

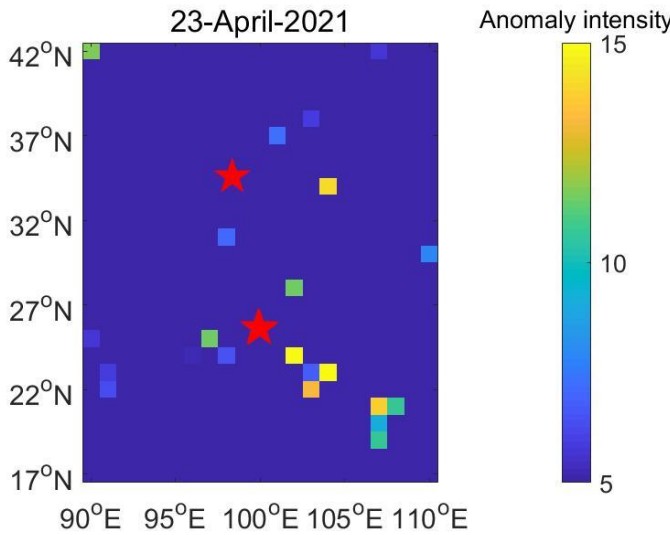

**Figure 17.** The relative power spectrum on 23 April 2021.

## 4. Conclusions

In this paper, the LSTM model is used to predict the TEC time series, and the predicted values are regarded as the background field of TEC. Then, the relative power spectrum method is used to extract the possible pre-seismic anomalies from the deviation of the observed value from the background field. The best combination of the intensity threshold, duration threshold, area threshold, predicted time window and predicted radius were searched in this paper. An algorithm with a low miss rate could be effective to predict a reasonable number of earthquakes. Although the high prediction range can improve the correlation rate and decrease the miss rate, it increases the spatiotemporal occupancy, which is not beneficial to improving the probability gain. In addition, the high spatiotemporal occupancy may cause unnecessary economic or investment losses and public panic. In summary, a creditable algorithm for earthquake prediction needs appropriate parameters with the comprehensive consideration of the correlation rate, the miss rate, and the probability gain. With the parameters groups selected in this paper, the probability gain is greater than 1.5 and the highest is 1.91. It shows that it is feasible to use TEC anomaly to reduce the uncertainty of prediction effectively. For group 1 in Table, the correlation rate is low (10%)

with a miss rate of 40% and a probability gain of 1.91. This means that the method has great limitations in earthquake prediction because the correlation rate is not high enough.

Taking two mainshocks that occurred on 21 May 2021 as an example, the high values of the relative power spectrum appeared many times before the earthquake near the two epicenters. Unexpectedly, the Yunnan earthquake, though smaller than the Qinghai earthquake, had more anomalies, and the abnormal phenomenon lasted longer. The reason for this phenomenon needs further investigation. Three days before the two earthquakes, there were anomalies near the two epicenters at the same time, and the intensity of the anomaly near the Qinghai epicenter was higher than that near the Yunnan epicenter.

The TEC anomalies detected in this paper show their potential for earthquake prediction. However, it is difficult to identify the earthquake precursor and guarantee a robust prediction by using a single type of remote sensing data with missing on-site measurements. Meanwhile, it is also not enough to study the causes and laws of earthquake precursors. Only by observing and analyzing multiple parameters of the atmosphere and the Earth's surface can we understand the development process of anomalies before earthquakes. We also confirm that the energy flowing in the lithosphere–atmosphere–ionosphere coupling system can be indirectly observed by the remote sensing-only method. Ouzounov et al. analyzed the pre-seismic anomalies of multiple parameters, including the TEC, the outgoing longwave radiation, the very low-frequency (VLF/LF) signals, the radon, the atmospheric chemical potential, and the air temperature [46]. Further research may need multi-source data including on-site measurements for earthquake prediction.

**Author Contributions:** Conceptualization, H.K. and M.B.-K.; methodology, F.C.; investigation, Y.Y.; writing—original draft preparation, Y.Y.; writing—review and editing, Y.C.; visualization, Y.C.; supervision, G.C.; funding acquisition, F.C. All authors have read and agreed to the published version of the manuscript.

**Funding:** This research received no external funding.

**Data Availability Statement:** Not applicable.

**Conflicts of Interest:** The authors declare no conflict of interest. The funders had no role in the design of the study; in the collection, analyses, or interpretation of data; in the writing of the manuscript; or in the decision to publish the results.

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
