# Peer review of "TEC Anomalies Detection for Qinghai and Yunnan Earthquakes on 21 May 2021"

_remotesensing, doi:10.3390/rs14174152_

Round 1

Reviewer 1 Report

Comments and recommendations for authors

This manuscript is well written. The structure and explanation are clear and can benefit readers as the flow is understandable. Here are my minor recommendations for the authors' action/consideration;

1.       Define m in line 116.

2.       Some sentences order need to be revised. For example, in line 139, the authors suddenly mentioned that ‘M is the length of the data’. This sentence could come out sprightly after the earlier equation.

3.       Incorrect the usage of ‘.’ and ‘,’ as in lines 130 and 141. The correct usage should be;

The fully connected layer could transform the output of the LSTM layer into the same dimension as the label, as shown in Equation (10); [23]

?̂? = ??ℎ? + ?? .

Authors are advised to revise the manuscript and make necessary changes.

4. Equations 19 and 20, and the description of their parameters, should appear on line 179, i.e, after mentioning them in line 178.

5.       Line 190, RSM E should be written as RSME.

6.       Some figures can be combined. For instance, Figure 5 and Figure 6 can be combined in 1 figure (right and left panel) for better visualization and comparison. Alternatively, Figure 6 and Figure 7 also can be combined.

Author Response

Thank you for your comments. I have revised my manuscript according to your suggestions. The response has been attached.

Reviewer 2 Report

I read the paper of Yue et al, about TEC anomalies before earthquakes in China in May 2021. The paper globally is well done. Furhtermore the Figure 12 is convincing that the authors were able to identify some pre-earthqaukes anomalous TEC signals that drop to standard values just few days after the seismic events occurrence. Their value shows strong anomalies about one month before the occurrence of the two earthquakes (M7.4 Qinghai and M6.4 Yunnan on 21 May 2021). Before this time the value are closer to zero, except some other values that in my point of view could be due to other pre-earthquake effects or other geophysics phenomena that could also have caused some TEC variations. The authors tested statistically by “confusion matrix” (even if they did not use this term) method that is the proper way to test a candidate of earthquake precursor. Despite this, I think the authors in this test confused some quantities and I ask them to revise deeply this section. Furthermore you must remove aftershock and foreshock from the earthqatquake catalogue as I point below in bullet points. The description of the literature can be improved especially in the introduction and I tried to provide you some examples of papers, but you can also search for your own surely.

Important questions:

l  Line 175. Do you consider any depth for the earthquakes? I would suggest to consider only shallow earthquakes at a maximum depth of 50 km or 60 km.

l  There is a confusion in the paper between true positive rate (called in the paper PPV) and hit rate that is called true positive rate (TPR). In particular:

o   Lines 175-176. In this sentence I found what was not so clear for me in the abstract. I suggest calling this quantity “true positive rate” TPR.

o   Lines 175-176. I suggest calling this quantity “true positive rate” TPR.

o   Line 176-177. This is the Hit rate not true positive rate.

o   Revise the abstract consequently.

l  You can find a description of these quantities for example here (you can also add the citation):

o   Fawcett, T. An Introduction to ROC analysis. Pattern Recogn. Lett. 2006, 27, 861–874. https://doi.org/10.1016/j.patrec.2005.10.010

l  Figure 5 and 6. You must color grey or white the boxes that don’t have a dataset, in fact, a blue value would correspond to an optimum model (low error) but zero means that there is no model. So You must use some graphic characteristic to mask these pixels (for example grey, white or some hatch pattern)

l  Figure 9. The Error is in TECU? You must specify the unit of measure (kp is okay without unit, Dst is in nT). An error of some decades of TECU seem quite large for me... please check or explain.

l  The considered earthquakes require some fundamental considerations, in fact, an event that occurs just before or after a big one is a foreshock or aftershock and you must count only the mainshocks. So considering your table 4. I see two mainshoks: Yunnan EQ M6.4 and Qinghai M7.4, the others are foreshocks and aftershock and I warmly suggest you to disregard them from computations. I mean table 4 can remain as you did, with another column where you specify if the EQ is a mainshock, aftershock or foreshock (even if this is not a universal definition, but in your case, I don’t see any criticality to define aftershock; mainshock; aftershock; aftershock; mainshock; foreshock for your 5 events. Then you must do again your statistics considering only the mainshocks. I’m sorry this could be some work for you, but this is the right way to evaluate the success of your method otherwise we have some overcount.

Specific points:

l  Line 20. What do you mean by “positive predicted value”? I think to understand reading the whole manuscript any way the abstract must be comprehensible also without reading the manuscript so I suggest replacing the terms.

l  The reference must be before the dot of sentence not after, please revise all. For example at line 32 “...earth’s surface and atmosphere [1].”

l  Lines 34-36. I would suggest to cite some possible Lithosphere-Atmosphere-Ionosphere coupling models and I agree that they are different and I principle it’s difficult to determine a pre-seismic anomaly really related to the seismic event:

o   Pulinets S, Ouzounov, D., 2011. LithosphereAtmosphere ionosphere coupling (LAIC) modelan unied concept for earthquake precursors validation, J. Asian Earth Sci, 41(4–5), 371–382. https://doi.org/10.1016/j.jseaes.2010.03.005

o   Kuo, C. L., Lee, L. C. and Huba. J.D., 2014. An improved coupling model for the lithosphere-atmosphere-ionosphere system. J. Geophys. Res. Space Physics, 119, 3189-3205, DOI: 10.1002/2013JA019392

o   Freund, F., Ouillon, G., Scoville, J., Sornette, D., 2021. Earthquake precursors in the light of peroxy defects theory: Critical review of systematic observations. Eur. Phys. J. Spec. Top. 230, 7–46. https://doi.org/10.1140/epjst/e2020-000243-x

o   Liperovsky, V.A.; Pokhotelov, O.A.; Meister, C., V.; Liperovskaya, E.V., 2008. Physical models of coupling in the lithosphere-atmosphere-ionosphere system before earthquakes. Geomagn. Aeron., 48, 795–806. https://doi.org/10.1134/s0016793208060133

l  Line 42. For CO2 anomalies I suggest to cite this paper that found interesting pre and co seismic emission of CO2:

o   Chiodini, G., Cardellini, C., Di Luccio, F., Selva, J., Frondini, F., Caliro, S., Rosiello, A., Beddini, G., Ventura, G. Correlation between tectonic CO2 Earth degassing and seismicity is revealed by a 10-year record in the Apennines, Italy. Science Advances, 2020, 6, 35, doi:10.1126/sciadv.abc2938.

l  Lines 43-50. I warmly suggest supporting these sentences with some literature. I totally agree with their content, but maybe some paper like this one can support more:

o   Spogli, L., Alfonsi, L., De Franceschi, G., Romano, V., Aquino, M. H. O., and Dodson, A.: Climatology of GPS ionospheric scintillations over high and mid-latitude European regions, Ann. Geophys., 27, 3429–3437, https://doi.org/10.5194/angeo-27-3429-2009, 2009.

l  Line 50. I think you can delete “phenomenon”. So, you can say “anomalies” or “anomalous phenomena”.

l  Line 52, please remove “seismic activity”. This is not something that affect your reserch goal but this is your goal.

l  Line 357 (reference 8) there is a “[C]//” after the title that can be removed. Also “Copernicus GmbH” I don’t think is necessary.

l  Lines 55-58. From this author there are two recent publications on TEC and LSTM neural Network applied to TEC from Swarm satellite:

o   Akhoondzadeh, M.; De Santis, A.; Marchetti, D.; Shen, X. Swarm-TEC satellite measurements as a potential earthquake precursor together with other Swarm and CSES data: The case of Mw 7.6 2019 Papua New Guinea seismic event. Front. Earth Sci. 2022, 10, 820189.

o   Akhoondzadeh, M.; De Santis, A.; Marchetti, D.; Wang, T. Developing a Deep Learning-Based Detector of Magnetic, Ne, Te and TEC Anomalies from Swarm Satellites: The Case of Mw 7.1 2021 Japan Earthquake. Remote Sens. 2022, 14, 1582.

l  Line 61. Maybe instead of “including” the authors can use “in terms of” but this is at your discretion

l  Lines 65-66. Please anticipate the reference [14] at the end of the first sentence.

l  Line 68. Maybe it’s better “previous data” instead of “prior data”.

l  Line 72. Even if the concept is right, I suggest to replace “unimportant signal” with “background signal”

l  Line 72. Okay, but maybe it’s better to start the sentence with “At the knowledge of the authors, there is no....”

l  Line 77-78. I would suggest to consider also the following statistical work on pre-EQ TEC anomalies in Japan:

o   Kon, S., Nishihashi, M., Hattori, K.. Ionospheric anomalies possibly associated with M P6.0 earthquakes in the Japan area during 1998–2010: Case studies and statistical study. Journal of Asian Earth Sciences 41 (2011) 410–420.

l  Line 79. I suggest to replace ”an earthquake after TEC anomalies” with “an earthquake occurrence after anomaly observation.”

l  Line 81 “seasonal change”

l  Lines 89, 93, 96 and 97, please check this is the degree symbol (it should be upper case).

l  Line 89 replace the dot after minutes with a space.

l  Line 90 I suggest “a new time series.”

l  Line 91. This earthquake is known also as the Madoi earthquake. I suggest the author to insert a sentence that says that is known also as “Madoi earthquake” see, for example:

o   Jing, F.; Zhang, L.; Singh, R.P. Pronounced Changes in Thermal Signals Associated with the Madoi (China) M 7.3 Earthquake from Passive Microwave and Infrared Satellite Data. Remote Sens. 2022, 14, 2539. https://doi.org/10.3390/rs14112539

l  Line 93. I think it’s better to use the past tense: “was located” furthermore maybe you can add some few information about the location that was in a high-mountain context luckily with few populations.

l  Line 97-98 I suggest to replace the sentence with “31 earthquakes with a magnitude of 5 or greater occurred in this region from 2020 to 2021.”

l  Line 99. You can replace “because of” with “due to”

l  Line 100. Please remove “There are” otherwise the sentence is wrong in English.

l  Line 111 – 112. Please remove the first “dataset”:  the training and test datasets”

l  Lines 117-118. I suggest replacing with “using the data of previously 72 hours, 48 hours and 24 hours.”

l  Line 130. If you can add a peer-review paper citation in addition to the one of ArXiv would be better.

l  Lines 148-150. I would suggest reformulating the sentence in this way: “these parameters are determined by the previous algorithm...”

l  Lines 153-154. I suggest shifting TEC close to the power spectrum: “the point with a high value of TEC relative power spectrum may be regarded as the possible pre-seismic anomaly”

l  Line 155. You can remove “value” before “threshold”.

l  Figure 4. Please insert also in the caption the meaning of the colours in the scheme

l  Eq. 17 and line 166. I suggest to change the letter that you used for a1, a2, b1 and b2. Firstly, use the same as before so alpha and beta for latitude and longitude. You can replace the 1 and 2 with m and M for minimum and Maximum or other letters, like the W and E for longitude that stay for West and East and S and N for beta (South and North), so αW, αE, βS, and βN.

l  Lines 168-171. This sentence is in bold. Is it a mistake or not?

l  Lines 175-176. In this sentence I found what was not so clear for me in the abstract. I suggest calling this quantity “true positive rate” TPR.

l  Line 179. This quantity in some papers like Shcherbakov et al., 2010 is called “alarm rate” and indicated with Greek letter tau. You can keep the name you give but cite this possible alternative.

o   Shcherbakov, R.; Turcotte, D.L.; Rundle, J.B.; Tiampo, K.F.; Holliday, J.R. Forecasting the Locations of Future Large Earthquakes: An Analysis and Verification. Pure Appl. Geophys. 2010, 167, 743–749.

l  Line 180. Sorry this sentence is not immediately comprehensible. If I understand well what you want to say, I suggest changing “determine better R and W” in “determine a better combination of R and W” or “determine the best combination of R and W”

l  Line 190. Please pay attention to a space between RMS and E

l  Figure 5 and or 6. I suggest to add an indication (like a cross) over the 3 points that you study after and reported in Table 1 and Figure 8.

l  Line 206. I suggest replacing “obviously.” with “, as expected.”

l  Figure 8 caption. You can avoid repeating data: “The observation and prediction data” and add “s” at “positions”. Maybe you can explain something more in the caption, for example, the time span, algorithm (LSTM) etc...

l  Line 209. Please replace “storms” by “activity” in fact you have a geomagnetic storm when there is high geomagnetic activity but you properly plot all the time not only geomagnetic storm time.

l  Line 211. For kp I agree, for Dst I don’t fully agree with your sentence. The geomagnetic activity is quiet as Dst is closer to zero, in fact also an increase above 20 nT is expected in the Sudden Commencement Storm, that is the impact of a geomagnetic storm on the Earth’s environment. So, revise the sentence accordingly.

l  Line 224-225 This is not true, all the parameters contribute to define an anomaly a precursor or not, in fact, a higher or lower threshold on the TEC value will also influence the classification of the anomaly.

l  Table 2. Please insert a space after comma of each quantity.

l  Table 2 do not use in the title the “/” indication for the unit of measurements that is only in Chinese like W / needs to be (W, days). Also (R, km)

l  Line 233 I think it’s better “It means” instead of “That is”

l  Table 3. I saw you ordered the table for decreasing gain. I fully agree with your choice but please specify it in the caption or text.

l  Table 4. Please specify that time is UT. For magnitude 5 event write 5.0 please

l  Figures 10 and 11. For the vertical axis use the brackets for the unit of measurement (km). When you edit these figures you can replace for minus “-“ with “" in horizontal axis for negatives days, otherwise it will be required in the proofreading stage in case of paper acceptance.

l  Line 270. I think this is not Yunnan but Qinghai earthquake in Figure 11. Please check.

l  Line 270. Please remove “in Table 4” as it is not necessary to refer again to table 4 that is misleading in this part of the text that must be more related to anomalies.

l  Line 271. Please replace by anomalies: “there are anomalies” abnormal is an adjective or you say abnormal points or anomalies (this second I think is better).

l  Line 281. I would suggest adding one month before the earthquake. This information is important and can confirm your value of 28 days for the best length of window you found in table 3 (i.e. the highest Gain >2). Please add such consideration if you agree.

l  Line 284. This is wonderful from my point of view because it’s in accordance with the magnitude of the two events. If you agree add the consideration.

l  Line 298. Please specify which parameters and maybe it’s better to say that you search for their best combination instead of say “superior result” that is not so scientific, sorry...

l  Lines 319-322. I completely agree with your sentence/conclusion and I would suggest you an example of combination of such parameters with TEC (radon, temperature Outgoing Wave radiation) in the occasion of Nepal 2015 earthqauke that happen in a similar tectonic context and it is another example of continental earthqauke. You can add the citation if you agree:

o   Ouzounov D, Pulinets S, Davidenko D, Rozhnoi A, Solovieva M, Fedun V et al. 2021. Transient Effects in Atmosphere and Ionosphere Preceding the 2015 M7.8 and M7.3 Gorkha–Nepal Earthquakes. Front. Earth Sci. 9:757358. doi: 10.3389/feart.2021.757358

l  Some references present [J] that I don’t know what is it, please check.

Author Response

(The authors gave the same response as above.)

Round 2

Reviewer 2 Report

Dear authors, thank you very much for the reply and revised version of the manuscript. I think the manuscript has been improved with respect to the first submission both in terms of research content and presentation of the methods and results. Despite this, there are still some points that can/must be improved.

Checking the new version, I found a crucial point on the interpretation of statistical quantities (missing rate and gain) that must be addressed before an eventual acceptance of the paper and several minor points that I ask you please to check/further revise.

There is a major point that I don’t agree with, sorry: you stay in the paper (line 801-802) and conclusion (1032-1033) that the low miss rate assures that the anomalies are related to the earthquakes and not just the occasional. That statistical point is not controlled by the missing rate but by the Gain that you calculated. In fact, you can correctly predict a few earthquakes founding only anomalies related to this small percentage of earthquakes. What about you discovered that maybe the smaller part of earthquakes that you predict are the higher magnitude (so the more likely to produce ionospheric anomalies)? Or they could be the shallower earthquakes. The important to be sure that the anomalies are not occasional is the Gain which is for definition the number of times that you are above a random prediction (so causal relation). On the other hand, a low missing rate assures that the prediction system is useful but it is not related to casualty. So, please revise this statistical section properly.

Minor points:

l  Line 82. Please check the spell of “model” not “modal”

l  Line 111. For me it’s okay you don’t use ROC to validate your study. I check the reference, but I would prefer you write “ROC could not be suitable for earthquake prediction” as several authors also among the other reference of your paper still use ROC as a valid tool. So please, just change with the conditional verb form.

l  Line 236. You cannot start a sentence with “That”. I suggest you replace it by “Such event”

l  Line 277 (Caption of Figure2): Please correct “epicenter” with “hypocenter” (or “earthquake”), in fact, the epicenter is the projection of the hypocenter on the Earth’s surface and so for definition depth is “zero”.

l  Line 247. Following the precedent comment, please remove “epicenter” you can say just “their depth”

l  Line 247. You can replace in “the deepest earthquake occurred at a depth of about 100 km.”

l  Line 248. Also here please remove “epicenter”, just “whose depth is no more...”

l  Lines 249-252. Even if the sentence is right as a description of the event occurred on 21 May 2021 in China, I would like you inserted a scientific comment otherwise I don’t found useful the sentence. I think the author would like to say that “Four M5+ earthquakes happened in Yunnan within 5 hours before the M7.4 Qinghai earthquake, suggesting a possible seismic trigger of this larger event. In addition, one foreshock with a magnitude of 5.1 happened in Qinghai about eight hours after the M7.4 Qinghai earthquake.”

l  Line 253. You can write “mainshocks” without space. Some English grammar spell check indicate as a mistakes, but scientifically the word is right.

l  Line 256. I suggest to replace the title “The information list of the earthquakes occurred in the study area from 2020 to 2021”.

l  Line 440. Thank you to have revised this part. I would suggest to add in bracket (confusion matrix) as this is the proper name of this statistical analysis. “According to the statistical analysis (confusion matrix) of TIR anomalies...”

l  Line 562. Please add the close bracket after (ν) symbol

l  Lines 572-573. Please rephrase this sentence. I suggest for example: “The prediction performance could be improved by selecting appropriate R and W.”

l  Line 665. Please add TECu at the end of the sentence. It’s important in a scientific paper to include the unit of measurements in the text and graphs (in this second submission the graphs are okay for unit of measurements).

l  Figure 10. Thank you for the modification, now the figure for me is okay. Just please spell the time in English and not Chinese “”, “”... Thank you also for the integration of Figure 11 with the histogram distribution of Errors value that clearly shows that the width of the distribution is contained and in particular, is about 2 TECu at FWHM, that is okay for me.

l  Line 775 The sentence is right in English but I would suggest to replace “enough” with “a reasonable number of” that is more scientific soundness.

l  Reference 23 is already listed as reference 14, so please at line 100 you should change in “Akhoondzadeh et al. [14, 23]” and changing all the following reference number according...

l  Line 1214. Please pay attention to separate family name and first name of the first author of the paper, I think is “Chen Wenkay”

Author Response

Thank you for your review. The response has been attached. Please see the attachment.
